# Opportunistic detection of type 2 diabetes using deep learning from frontal chest radiographs

Ayis Pyrros[1,2,20] ✉, Stephen M. Borstelmann[3,20], Ramana Mantravadi[4], Zachary Zaiman[5], Kaesha Thomas[5], Brandon Price [6], Eugene Greenstein[7], Nasir Siddiqui[1], Melinda Willis[1], Ihar Shulhan [8], John Hines-Shah[1], Jeanne M. Horowitz[9], Paul Nikolaidis[9], Matthew P. Lungren[10,11,12], Jorge Mario Rodríguez-Fernández [13], Judy Wawira Gichoya [5], Sanmi Koyejo [14], Adam E Flanders[15], Nishith Khandwala[16], Amit Gupta[17], John W. Garrett [18], Joseph Paul Cohen[11], Brian T. Layden[19], Perry J. Pickhardt[18] & William Galanter[19]

Deep learning (DL) models can harness electronic health records (EHRs) to predict diseases and extract radiologic findings for diagnosis. With ambulatory chest radiographs (CXRs) frequently ordered, we investigated detecting type 2 diabetes (T2D) by combining radiographic and EHR data using a DL model. Our model, developed from 271,065 CXRs and 160,244 patients, was tested on a prospective dataset of 9,943 CXRs. Here we show the model effectively detected T2D with a ROC AUC of 0.84 and a 16% prevalence. The algorithm flagged 1,381 cases (14%) as suspicious for T2D. External validation at a distinct institution yielded a ROC AUC of 0.77, with 5% of patients subsequently diagnosed with T2D. Explainable AI techniques revealed correlations between specific adiposity measures and high predictivity, suggesting CXRs' potential for enhanced T2D screening.

The prevalence of diabetes mellitus in the US population is approximately 10%, with the vast majority of cases being type 2 diabetes mellitus (T2D)[1,2]. The cost of US diabetes care was estimated to be $327 billion in 2017, primarily in patients over 65 years, significantly contributing to rising Medicare costs[3]. Similarly, the International Diabetes Federation estimates that 415 million people worldwide had diabetes in 2015, with potentially half of patients undiagnosed and at greater risk of complications[4]. Prediabetes is antecedent to T2D, typically for 9–12 years, with current common screening tests measuring fasting blood glucose (FBG) and/or hemoglobin A1C (HbA1c) levels[5]. The current recommendations of the American Diabetes Association and US Preventive Services Task Force (USPSTF) advise opportunistic

[1]Duly Health and Care, Department of Radiology, Downers Grove, IL, USA. [2]Department of Biomedical and Health Information Sciences, University of Illinois Chicago, Chicago, IL, USA. [3]Department of Radiology, University of Central Florida, Orlando, FL, USA. [4]Brainnet, Inc., West Harrison, NY, USA. [5]Department of Radiology, Emory University, Atlanta, GA, USA. [6]Department of Radiology, Florida State University, Tallahassee, FL, USA. [7]Department of Cardiology, Duly Health and Care, Downers Grove, IL, USA. [8]EPAM, Inc, Newtown, PA, USA. [9]Department of Radiology, Northwestern University, Chicago, IL, USA. [10]Department of Biomedical and Health Information Sciences, UCSF, San Francisco, CA, USA. [11]Center for Artificial Intelligence in Medicine, Stanford University, Stanford, CA, USA. [12] Microsoft, Microsoft Corporation, Redmond, USA. [13] Department of Neurology, The University of Texas Medical Branch, Galveston, TX, USA. [14]Department of Computer Science, Stanford University, Stanford, CA, USA. [15]Department of Radiology, Thomas Jefferson University, Philadelphia, PA, USA. [16]Bunkerhill, Palo Alto, CA, USA. [17]Department of Radiology, University Hospitals Cleveland Medical Center, Cleveland, OH, USA. [18]Department of Radiology, University of Wisconsin, Madison, WI, USA. [19]Department of Medicine, University of Illinois Chicago, Chicago, IL, USA. [20]These authors contributed equally: Ayis Pyrros, Stephen M. Borstelmann. ✉e-mail: ayis@uic.edu

3-year screening[2,5] for prediabetes and T2D in adults aged 35 to 70 years who are overweight or obese[2].

In the US, the prevalence of diagnosed diabetes has increased from 4.6% to 11.7%, but the prevalence of persistent undiagnosed diabetes (from 2.23% to 2.54%) and confirmed undiagnosed diabetes (from 1.10% to 1.23%) remains unchanged when comparing 1988–1994 and 2017 to March 2020[6]. Undiagnosed diabetes is more prevalent in older and obese adults, racial/ethnic minorities, and those with limited access to healthcare[6]. These underserved populations with limited healthcare access could offer an additional opportunity for detection, even though chest radiographs (CXRs) are not the primary method for detecting diabetes.

CXRs remain one of the most common radiologic exams[7,8], with over 26 million radiographs reimbursed by Medicare in 2017. As chest radiography is a common procedure in the US population, CXRs could be readily leveraged to detect undiagnosed diseases. Body mass index (BMI) is a poor predictor of T2D with many inherent flaws[9,10]. The particular fat depot, such as visceral fat, is an important risk factor for T2D[11] and can be quantified and used as an independent predictor[12]. However, its use in clinical practice is limited[13]. Additionally, the thin-fat phenotype found in Asian Indians has been increasingly recognized[14], typically not presenting with obvious adiposity, making its clinical detection more difficult. As BMI remains the primary clinical metric, despite its limitations[15], other indicators or predictors of T2D would be useful.

Utilizing deep learning (DL) methodology could create a revolution in disease detection through advanced biomarkers[16,17], allowing for the implementation of population-based health initiatives based on existing data in the electronic health record (EHR). Research has already demonstrated how DL with abdominal computed tomography imaging can detect numerous biomarkers predictive of, for example, metabolic syndrome in asymptomatic adults[18]. Likewise, DL with chest radiography has been shown to predict future healthcare expenses, health disparities, and multiple comorbidities[19–21]. Because of limitations around BMI, we aimed to explore the use of a multitask DL model to detect prevalent T2D from ambulatory frontal CXRs in a large clinical dataset.

In this work, we demonstrate the potential of utilizing deep learning models to detect type 2 diabetes (T2D) by combining ambulatory chest radiographic and electronic health record (EHR)

data. Our model was developed from a large clinical dataset of over 270,000 CXRs and 160,00 patients and tested on a prospective dataset of nearly 10,000 CXRs. The results showed that our model effectively detected T2D with a ROC AUC of 0.84 and a 16% prevalence, flagging 14% as suspicious for T2D. External validation at a geographically distant institution yielded a ROC AUC of 0.77, with 5% of patients subsequently diagnosed with T2D. Furthermore, using explainable AI techniques, we identified correlations between specific adiposity features and high predictivity, suggesting the potential of CXRs for enhanced T2D screening. These findings demonstrate the potential of DL models in harnessing demographic and administrative EHR data for disease discovery highlighting the potential for population-based health initiatives based on existing data.

## Results

### Dataset summary

Our model was developed from 271,065 CXRs (160,244 unique patients), sourced from 2010 to 2021 (the development cohort, our training dataset), which was prospectively evaluated on 9943 CXRs in 2022 (the prospective cohort, our test dataset) (Fig. 1). The original training dataset was further evaluated by k-fold cross-validation techniques (the retrospective internal validation dataset). We next externally validated with 5026 CXRs (Fig. 1) from a separate institution (the Emory cohort, our external validation dataset).

### Main analysis results

We developed a DL model using 11 years of data from 160,244 patients using their first ambulatory CXR to produce a prediction of the diagnosis of T2D (Supplementary Table 1). We also produced a logistic regression (LR) model that did not include any image information from the CXRs. The performance of the CXR DL model for the prediction of T2D in a separate test cohort of 9,943 patients with a CXR was 0.84 (95% confidence interval [CI]: 0.83, 0.85) compared with the LR model, which had an AUC of 0.79 (95% CI: 0.77, 0.80; P < 0.001 for comparison of the significance of the AUC difference between the two models.)

In subjects with poorly controlled T2D (defined as HbA1c ≥ 9% at any time for a patient) versus all others, the CXR DL predictor demonstrated similar performance (AUC = 0.85, 95% CI: 0.83, 0.86). In a subgroup analysis of subjects who meet the USPSTF criteria in

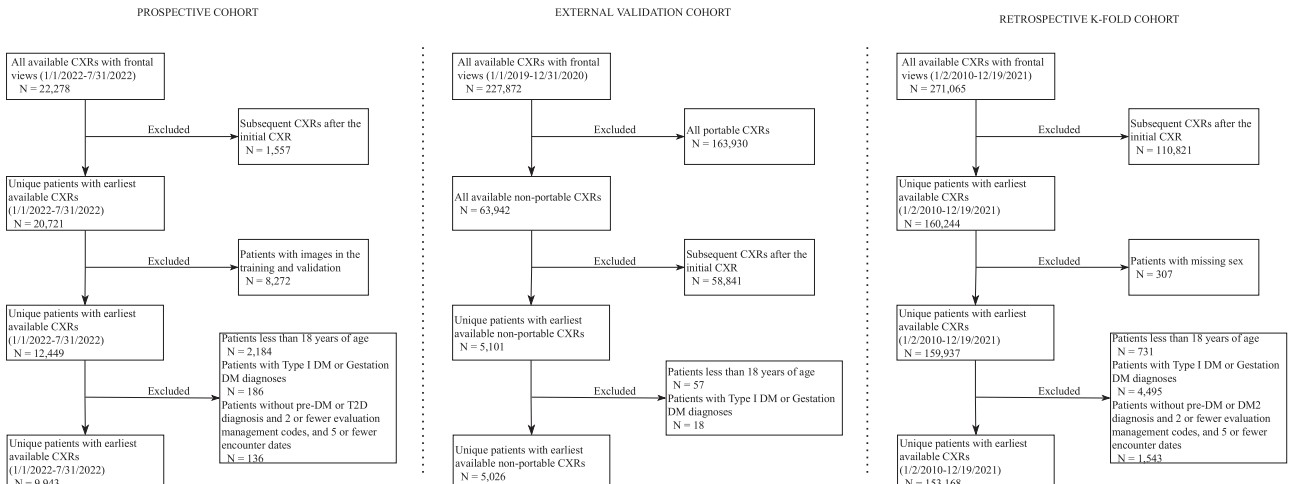

**Fig. 1 | Flowchart depicting the selection process for cases and controls from the prospective, external validation, and retrospective k-fold cohorts.** Exclusions were made sequentially based on ICD codes as presented in the flowchart (patients may have multiple exclusionary diagnoses but were counted only once according to their first exclusionary diagnosis). In the prospective and retrospective cohorts, patients with two or fewer claims for evaluation and management

codes (CPT codes 99202 to 99499) and five or fewer unique encounter dates were excluded, as they may have received care in another health system. Patients diagnosed with type 1 diabetes (ICD9: 250.x1, 250.x3; ICD10: E10.x) and gestational diabetes (ICD9: 648.80–648.84; ICD10: O24.4x) were also excluded from all cohorts as potential confounders. The final number represents unique patients with a single conventional frontal chest radiograph.

**Table 1 | Prospective cohort performance results compared by model**

| | Clinical logistic regression model[a] | | | | | | | | | | | CXR deep learning model | | | | | | | | | | | Deep learning with logistic regression model[a] | | | | | | | | | | DL contribution | |
|---|---|---|---|---|---|---|---|---|---|---|---|---|---|---|---|---|---|---|---|---|---|---|---|---|---|---|---|---|---|---|---|---|---|
| | N | N_total | Sens.[b] | Spec.[b] | NPV | PPV[b] | Prev.% | F1 | AUC | AUC CI | P[c] Value | N | N_total | Sens.[b] | Spec.[b] | NPV | PPV[b] | Prev.% | F1 | AUC | AUC CI | P[c] value | N | N_total | Sens.[b] | Spec.[b] | NPV | PPV[b] | Prev.% | F1 | AUC | AUC CI | OR (95% CI) | P[d] value |
| All T2D vs. All others | 1554 | 8126 | 0.79 | 0.65 | 0.94 | 0.3 | 16.1 | 0.43 | 0.79 | (0.77, 0.80) | 2.2 ×10⁻¹⁶ | 1561 | 8382 | 0.85 | 0.68 | 0.96 | 0.33 | 15.7 | 0.48 | 0.84 | (0.83, 0.85) | 0.16 | 1554 | 8126 | 0.78 | 0.76 | 0.95 | 0.38 | 16.1 | 0.51 | 0.85 | (0.84, 0.85) | 93 (66, 135) | 2 × 10⁻¹⁶ |
| Poorly controlled T2D vs. all others | 440 | 9240 | 0.76 | 0.61 | 0.98 | 0.08 | 4.5 | 0.15 | 0.74 | (0.72, 0.76) | 2.2 ×10⁻¹⁶ | 442 | 9501 | 0.84 | 0.74 | 0.99 | 0.13 | 4.5 | 0.22 | 0.85 | (0.83, 0.86) | 0.79 | 440 | 9240 | 0.86 | 0.71 | 0.99 | 0.13 | 4.5 | 0.22 | 0.85 | (0.83, 0.87) | 107 (67, 17) | 2 × 10⁻¹⁶ |
| T2D with BMI < 25, 35–70 vs no T2D | 79 | 1241 | 0.71 | 0.75 | 0.98 | 0.15 | 6 | 0.25 | 0.77 | (0.72, 0.82) | 5.8 × 10⁻⁸ | 79 | 1241 | 0.84 | 0.79 | 0.99 | 0.20 | 6 | 0.33 | 0.89 | (0.85, 0.93) | 0.22 | 79 | 1241 | 0.86 | 0.76 | 0.99 | 0.18 | 6 | 0.30 | 0.87 | (0.83, 0.92) | 6,837 (783, 59,722) | 1.4 × 10⁻¹⁵ |
| T2D with BMI ≥ 25, 35–70 vs no T2D | 885 | 3909 | 0.73 | 0.6 | 0.91 | 0.29 | 18.5 | 0.42 | 0.74 | (0.71, 0.75) | 2 ×10⁻¹³ | 885 | 3909 | 0.74 | 0.72 | 0.93 | 0.38 | 18.5 | 0.5 | 0.80 | (0.79, 0.82) | 0.03 | 885 | 3909 | 0.80 | 0.69 | 0.94 | 0.37 | 18.5 | 0.51 | 0.81 | (0.80, 0.83) | 86 (55,135) | 2 × 10⁻¹⁶ |

T2D type 2 diabetes, No T2D no diabetes, NPV negative predictive value, PPV positive predictive value, Sens sensitivity, Spec specificity.
[a]Logistic regression models all included age, sex, BMI, race/ethnicity, language preference, and SDI; differences in case counts between DL and LR models due to observations being deleted due to missingness in LR models.
[b]NPV, PPV, sensitivity, and specificity were calculated using Youden's index for an optimal threshold.
[c]P Value, AUC two-sided comparison using the method DeLong between the adjacent models. No modifications were implemented for multiple comparisons since the models were evaluated on a pairwise basis.
[d]P value the DL for predictor in a logistic regression model. all models, exponentiated coefficient.
Source data are provided as a Source Data file.

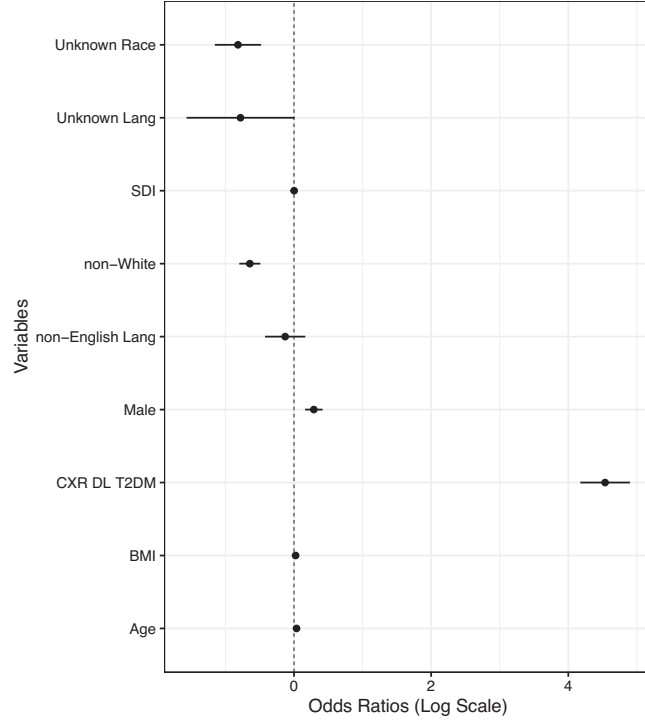

**Fig. 2 | Integration of clinical LR and CXR DL T2D predictors.** The combined clinical LR and CXR DL T2D predictor displays the odds ratios for diabetic patients (poorly controlled and controlled T2D, $n = 1554$) compared to patients without diabetes ($n = 8126$) using a logarithmic scale and 95% CI error bars. Self-reported race is expressed relative to White, and self-reported language preference is relative to English. The logarithmic transformation of odds ratios is used to enhance visualization. Source data are provided as a Source Data file.

screening[2] for T2D (BMI ≥ 25, age between 35 and 70 years), the CXR DL predictor had an AUC = 0.80 (95% CI: 0.79, 0.82), and in a cohort for BMI < 25 (ages between 35 and 70), the CXR DL model reached an AUC = 0.89 (95% CI: 0.85, 0.93). The CXR DL model consistently outperformed the clinical LR model at a significance level of <0.001. Full results are presented in Table 1.

To evaluate the importance of the CXR DL's prediction overall, we added it as an input into an LR model (DL with LR model). The CXR DL predictor contribution dominated the overall LR via its odds ratio (Fig. 2 and Table 1). AUC for the prediction of T2D improved vs. the clinical LR baseline model; however, it was not statistically significant: 0.85 (95% CI: 0.84, 0.85) versus 0.84 (95% confidence interval [CI]: 0.83, 0.85, $P = 0.16$). AUC also improved for the subset of patients who met USPSTF screening criteria (AUC = 0.81, 95% CI: 0.80, 0.83, $P = 0.03$), also included in Table 1.

As shown in Fig. 3, the DL model predictions for all subjects with T2D were significantly higher than those for subjects without T2D (median 0.29; interquartile range [IQR]: 0.15, 0.49 vs. median 0.04; IQR: 0.01, 0.14; $P < 0.001$; Fig. 3A). Subjects with poorly controlled T2D had higher scores (median 0.35; IQR: 0.20, 0.58) than subjects with controlled T2D (median 0.26; IQR: 0.13, 0.45) or no diabetes (median 0.04; IQR: 0.01, 0.14; $P < 0.001$; Fig. 3B).

In the prospective test cohort, among all ages, 1381 (14%) patients were identified by the model as high risk using Youden's Index[22] (threshold greater than 0.10) who did not have an HbA1c value or a diagnosis of T2D, representing potential screening opportunities. Of these, 147 would not have met the criteria for screening per the USPSTF[2] (BMI < 25 amongst all ages), with an additional 70 uncertain based on no available BMI.

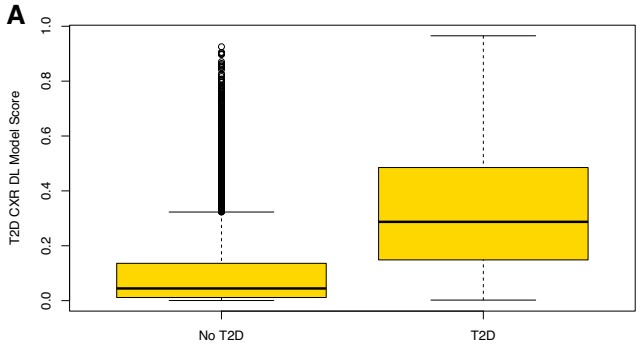

**Fig. 3 | CXR DL model scores for prospective cohort positive and negative for T2D.** Box and whisker plots show DL model scores (y-axis) for (**A**) no T2D versus T2D and (**B**) no T2D, controlled T2D, and poorly controlled T2D. Boxes represent the IQR (25–75%) with the median noted by the horizontal bar within each box, whiskers extend from the box to the minimum and maximum values within 1.5

times the IQR, with circles representing outliers in the distributions (n = 9943). The two-sided Wilcoxon rank sum test was used to assess the difference in T2D DL model score and T2D diagnosis ($P < 2.2 \times 10^{-16}$). The Kruskal–Wallis test was used to compare differences between the three groups, which were significant ($P < 2.2 \times 10^{-16}$). Source data are provided as a Source Data file.

### Retrospective (k-fold) validation results

Internal retrospective validation was performed as specified in the k-fold internal validation section in "Methods" (Supplementary Table 2). Results were similar to the prospective test set, with the DL model producing an AUC of 0.83 (95% CI: 0.82, 0.83) versus 0.84 in the prospective internal test cohort (Table 1 & Supplementary Table 3). In subjects with poorly controlled T2D versus all others, the DL predictor had an AUC = 0.82 (95% CI: 0.81, 0.82). In a cohort for which BMI was <25 the AUC = 0.83 (95% CI: 0.81, 0.84). In subjects who meet the USPSTF criteria in screening for T2D (BMI ≥ 25, age between 35 and 70 years), the DL model reached an AUC = 0.79 (95% CI: 0.79, 0.79). Results are tabulated in Supplementary Table 3.

**Table 2 | Prospective cohort: comparison of relevant characteristics per cohort**

| Patient characteristic | Total N = 9943 (%)[a] | | |
|---|---|---|---|
| | No T2D (N = 8382) (84.3%) | Controlled T2D (N = 1119) (11.3%) | Poorly Controlled T2D (N = 442) (4.4%) |
| Age, mean (SD), y | 51.9 (18.3) | 67.0 (12.7) | 60.1 (12.5) |
| Sex | | | |
| Women (%) | 4897 (58.4) | 546 (48.8) | 198 (44.8) |
| Men | 3485 (41.6) | 573 (51.2) | 244 (55.2) |
| Self-Reported Race/Ethnicity | | | |
| White, Non-Hispanic (%) | 5905 (70.8) | 727 (65) | 258 (58.4) |
| Black, Non-Hispanic | 513 (5.8) | 118 (10.5) | 46 (10.4) |
| Asian, Non-Hispanic | 522 (6) | 116 (10.4) | 37 (8.4) |
| Hispanic | 723 (8.7) | 85 (7.6) | 68 (15.4) |
| Other | 719 (2.1) | 73 (2.8) | 33 (7.5) |
| Other clinical variables | | | |
| BMI, mean (SD)[b] | 29.24 (6.98) | 32.26 (7.44) | 32.51 (7.60) |
| SDI, mean (SD)[c] | 25.91 (23.81) | 30.12 (26.86) | 30.06 (26.32) |
| HbA1c, mean[d, e] | 5.48 | 6.61[e] | 6.59[e] |

[a]Data are given as numbers (percentage) for each group, unless specified.
[b]Missing values = 310.
[c]Missing values = 2.
[d]Missing values = 5514.
[e]The mean HbA1c values include the most recent results, not just the times when the patient may have had T2D or poorly controlled T2D.
Source data are provided as a Source Data file.

### Incidence detection of T2D

In the 11-year retrospective k-fold cohort, 7409 (25%) of the 29,420 patients had a diagnosis of T2D after the initial CXR. The incidence rate of T2D in this population was 5.1 cases per 1000 people per year at risk (95% CI: 5.0, 5.3). Of these 7409 patients, 5292 (71%) had a DL prediction >0.10, again corresponding to the Youden's Index. Time-dependent ROC curves at 1 year (AUC = 0.80, 95% CI: 0.79, 0.81), 3 years (AUC = 0.79, 95% CI: 0.78, 0.80), 5 years (AUC = 0.79, 95% CI: 0.78, 0.80), and 10 years (AUC = 0.78, 95% CI: 0.77, 0.79) demonstrated similar performance over time (Supplementary Table 4). Using the CXR as the index date, the delay in diagnosis was an average of 1,057 days (SD ± 1005 days) and median 738; IQR: 256, 1,590.

### External validation results

The validation of the DL model was independently performed on an external dataset of ambulatory frontal CXRs at Emory from 2019 to 2020 (Supplementary Tables 5 through 7), and we observed an AUC of 0.77 (Supplementary Table 7). In this cohort, the incidence rate of T2D was 20.4 (95% CI: 18, 23) cases per 1000 person-years at risk, with 249 patients diagnosed with T2D after the initial CXR. Of the 249 patients, the model flagged 146 (59%) for potential earlier screening. Additional time-dependent ROCs were not performed on the external cohort, because of small sample size and short length of time.

### Demographics

Of the prospective test cohort's (Fig. 1) 9,943 patients, most had no T2D (n = 8,382; 84.3%) and some had controlled T2D (n = 1,119; 11.3%) or poorly controlled T2D (n = 442; 4.4%) (Table 2). Patients with T2D tended to be older than those in other cohorts at 67 years (SD: 12.7), and there was a predominance of female patients in the nondiabetic cohort and male patients in the controlled T2D and poorly controlled T2D cohorts. Regarding race/ethnicity, white Non-Hispanic individuals were prevalent in each subgroup, followed by Hispanic; Asian, Non-Hispanic; and Black, Non-Hispanic individuals. In addition, patients with poorly controlled T2D had higher BMI and social deprivation index (SDI). Demographics in the training dataset were similar and are shown in Supplementary Table 1.

### Model equity

Previous studies have shown that convolutional neural networks can easily learn self-reported race and other sensitive attributes[23,24]. Out of concern that spurious features related to these sensitive attributes could be contributing to this diabetes prediction model, a subgroup analysis was conducted and shown in Table 3. Subgroup analysis by race/ethnicity failed to achieve statistical significance (P > 0.05), suggestive of a lack of bias. There was a statistical difference in model

**Table 3 | Area under the receiver operating characteristic curve for evaluation of model equity**

| Characteristic | Cases | Controls | AUC (95% CI) (Delong) | Prevalence | NPV | PPV | Sensitivity | Specificity | F1 |
|---|---|---|---|---|---|---|---|---|---|
| Sex* | | | | | | | | | |
| Male | 817 | 3,485 | 0.83* (0.82, 0.84) | 19 | 0.93 | 0.41 | 0.76 | 0.74 | 0.53 |
| Female | 744 | 4,897 | 0.85 (0.84, 0.86) | 13 | 0.97 | 0.29 | 0.87 | 0.68 | 0.44 |
| Race/Ethnicity** | | | | | | | | | |
| Asian | 153 | 522 | 0.86 (0.83, 0.89) | 23 | 0.95 | 0.49 | 0.86 | 0.74 | 0.63 |
| Black | 164 | 513 | 0.80 (0.77, 0.84) | 24 | 0.89 | 0.47 | 0.72 | 0.74 | 0.57 |
| Hispanic | 153 | 723 | 0.84 (0.81, 0.87) | 18 | 0.96 | 0.37 | 0.88 | 0.68 | 0.52 |
| White | 985 | 5,905 | 0.84 (0.83, 0.86) | 14 | 0.97 | 0.3 | 0.87 | 0.66 | 0.45 |
| Unknown/Other | 106 | 719 | 0.84 (0.81, 0.88) | 13 | 0.97 | 0.32 | 0.82 | 0.74 | 0.46 |

*$P = 0.045$, Comparison of AUC employing a two-sided approach through the DeLong method.

**No significant differences ($P > 0.05$) in AUCs by race/ethnicity were observed after accounting for multiple pairwise comparisons using the two-sided DeLong method and applying the Holm–Bonferroni correction. NPV, PPV, sensitivity, and specificity were calculated using Youden's index for an optimal threshold. Source data are provided as a Source Data file.

performance by biological sex male vs female: 0.83 (95% CI: 0.82, 0.84) versus 0.85 (95% CI: 0.84, 0.86, $P = 0.045$).

### Model explainability
Occlusion maps were generated to display the basis for the model decision (Figs. 4 and 5), with image features predictive of T2D corresponding to the central chest, lower neck, upper abdomen, and axillary regions. In Fig. 5, we took random samples of occlusion maps from the internal and external cohorts to demonstrate that the same features were being used. In addition, we used an autoencoder and a latent shift to generate an animation ("gifsplanation") (Fig. 6), exaggerating and curtailing anatomic features used for prediction from a representative frontal radiograph[25]. This method also does not alter the model weights and demonstrates that central fat distribution (mediastinal, upper abdomen, and supraclavicular regions), as well as attenuation of the ribs and clavicle, drives the prediction for T2D. This animation can be directly viewed in Supplementary Movie 1 with multiple randomly selected examples. Additional analysis was done using the DL model prediction of HbA1c. For the retrospective cohort of patients between 2010 and 2021, we collected all HbA1c values within a ±30-day window of the CXRs ($n = 15,945$) and conducted a linear regression analysis between the HbA1c predicted by the DL model and the actual obtained HbA1c values (Fig. 7).

## Discussion
In this study, we developed a DL model that can accurately identify patients with T2D from routine frontal CXRs. Diabetes prediction in patients with T2D (including poorly controlled T2D) compared to subjects without diabetes showed an AUC of 0.84 (95% CI: 0.83, 0.85). There was performance improvement in the AUC when adding the CXR DL prediction to a LR model with only non-imaging variables. Overall, there is a complementary benefit of the ensemble DL model when predicting T2D in a large cohort.

Several important observations can be made from this study. First, the DL model performed well in detecting patients with prevalent T2D, with an AUC of 0.84 in patients with BMI ≥ 25 and an AUC of 0.89 in patients with BMI < 25. Second, the DL model increased discriminative performance and outperformed clinical LR models across multiple scenarios. Third, the prediction of poorly controlled T2D may offer more targeted interventions to higher-risk patients, such as enhanced screening. Fourth, the sensitivity (0.74) and specificity (0.72) of DL screening in a cohort with a BMI ≥ 25 exceeds the sensitivity and specificity of the USPSTF guidelines (0.45 and 0.69, respectively) from previously published results using the guidelines[26]. To our knowledge no other prospective observation study has tested the real-world performance of a DL model for T2D diagnosis based on CXRs.

Fourteen percent of patients in the prospective cohort who had not undergone HbA1c screening in this study were identified as potential screening opportunities despite significant contact with the healthcare system meeting or exceeding current standard of care approaches. Our retrospective cohort demonstrated that 7409 (25%) of the 29,420 patients had a diagnosis of T2D subsequent to the initial CXR over an 11-year period, representing a significant screening opportunity. Since CXRs are commonly obtained, they could be especially useful for opportunistic screening in patients who lack a primary care provider or only receive care through the emergency department. This would be especially impactful at population level, given that T2D has strong associations with social deprivation and social determinants of health. Additionally, with the increasing worldwide prevalence of T2D, patients determined by the algorithm to be at risk could be sent for HbA1c screening, with high-risk patients referred to a primary care provider via a pre-established order set, or could potentially undergo automatic screening via an ordered HbA1c. In fact, the model's ability to predict HbA1c from the CXR was limited, reinforcing the need for complementary HbA1c testing.

Explainability of the model was adjudicated by two different techniques: occlusion mapping and an autoencoder (gifsplanation). Both demonstrated central adiposity (mediastinal lipomatosis) as well as attenuation of the ribs and clavicles as predictive drivers. Studies have documented the importance of "upper body" or "abdominal" obesity as a determinant of insulin resistance, T2D, hypertension, dyslipidemia, and cardiovascular morbidity and mortality[27]. We believe the detection of this central mediastinal adiposity is why the DL model is able to predict T2D in patients with normal BMIs. Interestingly, the density and thickness of the bones is an additional feature that is used in prediction. The attenuation of the ribs and clavicle can increase the DL prediction, which could represent an increasing amount of adiposity obscuring the osseous structures, with other possibilities including age and diabetes-related osteoporosis[28]. This represents some of the opportunities available with DL to identify features not routinely reported but relevant nonetheless.

We believe that this CXR DL predictor is a useful tool to opportunistically augment conventional methodologies of diagnosing diabetes, by automated DL biomarker extraction, specifically in the context of identifying patients who could benefit from additional targeted screening. The advantage of this approach is that it demonstrates high performance and efficiency using data already collected for other purposes (i.e., ambulatory CXRs), providing an opportunity to extract valuable patient-specific data for use in care management. Both FBG and HbA1c levels are used for screening, but both have been found to under-diagnose T2D until its advanced stages[29]. A published model predicting poorly controlled T2D relied on extensive demographic information[30], but that same study also demonstrated social

|  | Low T2D DL Score | High T2D DL Score |
|---|---|---|

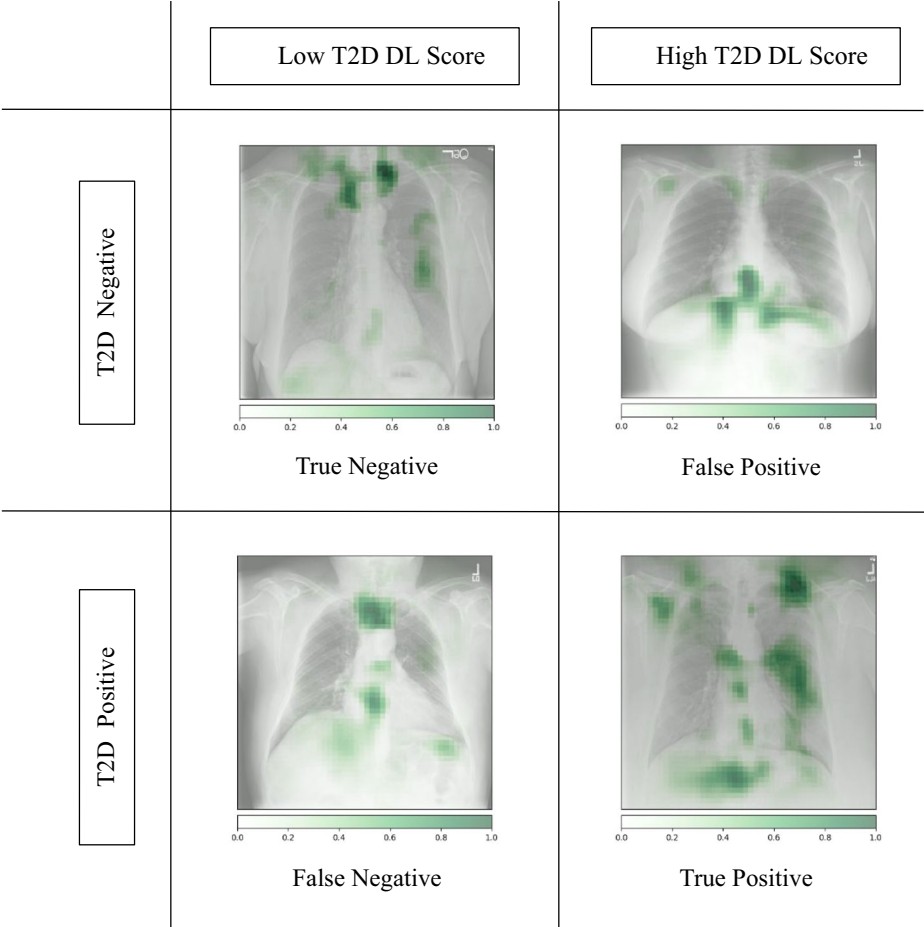

**Fig. 4 | Occlusion maps identifying key features in representative CXRs with high and low diagnostic scores.** Dark green pixels highlight significant features for model prediction, primarily associated with cardiomediastinal, upper abdominal, lower neck, and supraclavicular regions. Examples of CXRs with high and low diagnostic scores are presented.

determinants of health are only able to explain 16.9% of variation in poorly controlled diabetes, with such patients often having complex needs[31]. There is increasing evidence of genetic and epigenetic contributions for T2D, which are not clearly understood[32]. Other advanced models for the prediction of diabetes rely on extensive laboratory information[33] or whole-body magnetic resonance imaging[34], which limit the potential utility in clinical practice as an opportunistic tool due to high cost and rarity, as compared to the relative abundance and low cost of CXRs.

This DL approach to "opportunistic T2D screening" with medical imaging data obtained in routine care for other reasons is able to more granularly risk-stratify patients due to a continuous prediction with values 0–1, while current screening methods for T2D categorize as normal, prediabetes and T2D. In our study, it was not possible to directly compare the USPSTF criteria in our sample due to the lack of HbA1c data in many patients with a BMI over 25. Future work can be done to analyze and compare HbA1C values and the CXR DL prediction for differences in predictive power. It may be possible that patients with a normal HbA1C may be identified as having risk for future development of T2D by the CXR DL prediction.

Limitations include that we did not include FBG or other glucometry data in this study because of the inherent difficulties in confirming that patients were indeed fasting prior to obtaining the measurement. As the method was opportunist and retrospective, there was no HbA1C measurement done or available for many of the patients with a prediction, which may limit the accuracy of the training process. In addition, only ambulatory CXRs were utilized; no portable CXRs were used in development or testing. Likewise, the ambulatory nature of this study means no images with support devices like endotracheal tubes were used. In addition, there is current debate about the utility of early detection of T2D given the presumed length of time for complications to develop. On balance, this DL approach to "opportunistic T2D screening" with medical imaging data obtained in routine care for other reasons is able to more granularly risk-stratify patients, whereas current screening methods for T2D are unable to stratify patients. While an external validation was performed on data from an outside institution, it was neither extensive nor fine-tuned, and obtained results were generally as expected with the lack of calibration and smaller dataset. Further evaluation and fine-tuning would be warranted prior to widespread clinical implementation to ensure the model met standards for fairness and equity in DL models, as well as robustness to model drift and out-of-sample cases. This is, however, beyond the scope of this study. Finally, multi-year follow-up of the prospective test cohort is not yet available.

In summary, we developed a DL model that can accurately identify patients with T2D on routine CXRs. Using this model in population-level health efforts could potentially allow millions of patients with T2D to be identified earlier in the disease process. Starting preventive medication and implementing lifestyle changes could reduce the risk of associated DM complications like microvascular disease, kidney disease, heart disease, and stroke from existing CXR data acquired for other purposes. This DL model could also provide added value to routine radiologist interpretations via

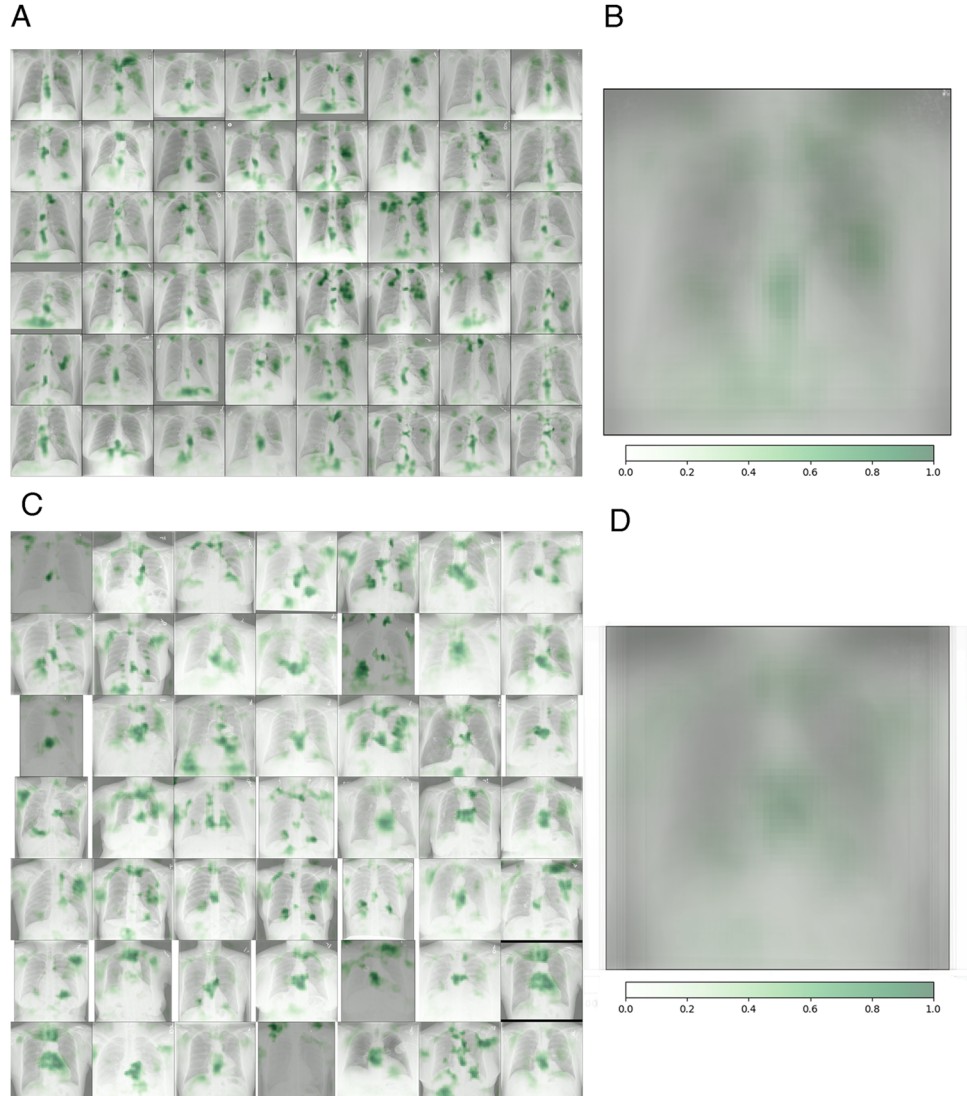

**Fig. 5 | Random sampling of occlusion maps from both internal and external cohorts. A** A random selection of 48 occlusion maps from the DL CXR prediction model, with green regions indicating crucial features. **B** A composite image of the 49 occlusion maps showcasing positive attribution to the central mediastinum, lower neck and supraclavicular fossae, upper abdomen, and ribs. **C** A random sampling from the external validation dataset. **D** An averaged map from the external data exhibiting a distribution similar to (**B**). Source data are provided as a Source Data file.

automated quantification and reporting on routine CXRs in practice, allowing patients to be flagged, automatic order sets for further testing to be triggered, and alerts to be sent to the responsible clinician or the patients themselves. Because the CXR is the most common imaging examination in the world for a wide variety of medical indications, this model could also be applied to large populations of CXRs to identify high-risk individuals and perform more accurate risk assessment, leading to significant advantages for population health efforts.

## Methods

### Ethics statement

This study received institutional review board (IRB) approval from both Edwards-Elmhurst (01-21-21_NHSR) and Emory (Chest x-ray - IRB0009197), ensuring adherence to all relevant ethical regulations. Given the retrospective anonymous nature of this research, a waiver of Health Insurance Portability and Accountability Act (HIPAA) authorization and informed consent was granted by both IRBs. No participant compensation was provided. Research reported in this publication is part of MIDRC (The Medical Imaging Data Resource Center) and was made possible by the National Institute of Biomedical Imaging and Bioengineering (NIBIB) of the National Institutes of Health under contracts 75N92020C00008 and 75N92020C00021. The authors controlled the data and information submitted for publication.

### Outcome ascertainment

The final outcome of this study was the diagnosis of T2D based on ICD9 or ICD10 diagnosis codes (ICD9: 250.02, 250.1, 250.22, 250.3, 250.32, 250.42, 250.5, 250.52, 250.62, 250.72, 250.8, 250.82, 20.9, 250.92, ICD10: E11.x, E12.x, E14.x, Z79.84, Z79.4) or an HbA1c value ≥6.5% at any time available in the EHR 1/1/2000 to 7/31/22. The outcome of poorly controlled T2D diabetes was defined as an HbA1c value ≥9% at any time point, as per prior studies[29,35,36]. The outcome was included over the 11-year cohort (2010-2021) and separately for the prospective cohort (2022).

### Setting

This study has three cohorts: a retrospective development/validation training cohort from 2010 to 2021; an internal prospective test cohort

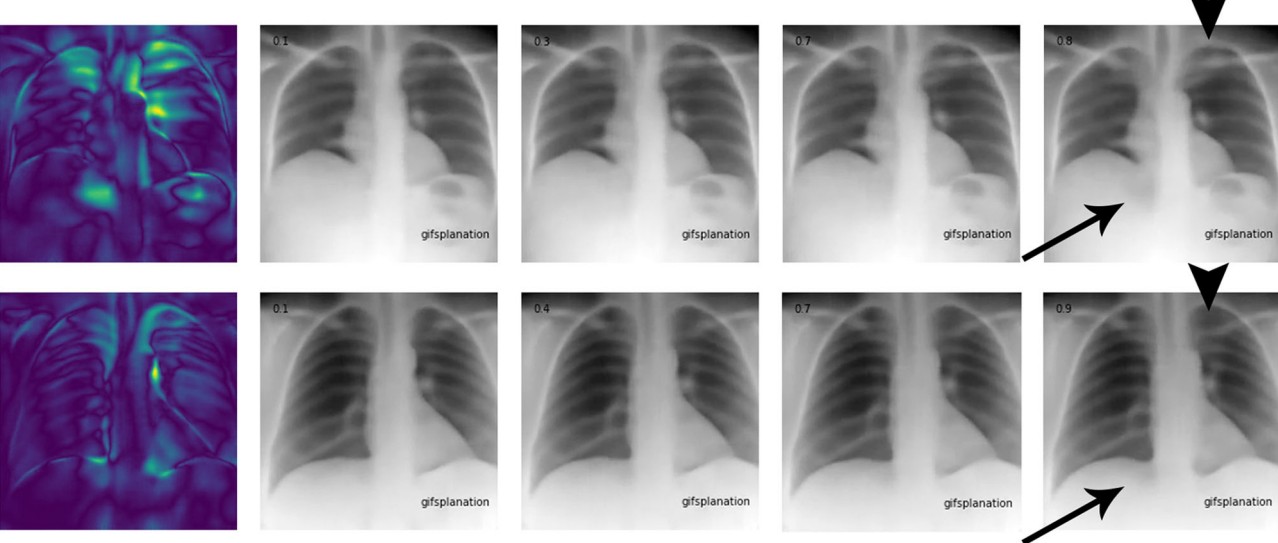

**Fig. 6 | Gifsplanation using Latent Feature Autoencoder.** The color heatmap highlights areas of change, with DL predictors progressively increasing along the horizontal axis in top and bottom rows. The change in central mediastinal adiposity is a primary driver. High predictive values (rightmost) include changes in upper abdominal fat (arrow) and supraclavicular and rib attenuation (arrowhead) which are intense upon the heatmap. The animation can be viewed as Supplementary Movie 1, which highlights the changes dynamically.

from January 1, 2022 to July 31, 2022; and an external test cohort obtained from Emory 2019 to 2020. Outpatient frontal CXRs were extracted from Duly Health and Care, a large multisite multispecialty medical group in the suburbs of Chicago, IL. Currently, Duly has over 150 sites with over 1000 providers, providing ambulatory care, with a subspecialty radiology group, maintaining its own PACS system and EHR. EHR data is shared between local hospitals, and extensively within one system, providing additional inpatient and outpatient clinical documentation. For the prospective portion of this study, CXRs were obtained at 28 geographically unique locations, utilizing 44 different units from manufacturers including GE, Philips, Toshiba, Konica, Summit, Quantum, Del Medical. The CXRs obtained were conventional, standard posteroanterior radiographs. External validation was performed utilizing a pre-existing dataset from Emory Hospital (EMX)[23].

## Model development

For the development training and validation dataset, we obtained 271,065 CXRs (unique cases) between 1/2/2010 and 12/19/2021 (mean age, 58.8 years ±17.5 [SD]; 55% women) (Fig. 1), with a total of 303,604 frontal CXR images. Demographics for the training data are provided in Supplementary Table 1. Negative controls, henceforth referred to as controls, were chosen to mimic the deployment environment, and because the DL model would be deployed on all adult CXRs, controls including all available CXRs in the date range above were utilized, along with cases of T2D. The labels for imaging training were based on ICD10 Hierarchical Condition Category (HCC) codes (2021 model 24) for six disease classes, including T2D, congestive heart failure, cardiac arrhythmias, morbid obesity, chronic obstructive pulmonary disease, and vascular disease[21]. Codes mapped to a category were binary encoded to 1 (True), and absent codes mapped to 0 (False), utilizing the most recent codes as of December 2021. Additional training data included BMI (kg/m²) and HbA1c closest to the obtained CXR, as well as patient age at the time of CXR. For model development, 218,758 CXR images were used for training, with 24,529 CXR images in validation (90%/10% split), and 60,317 used in testing.

We have previously published similar technical and hyperparameter details for a multitask DL model, which was externally validated at an urban hospitalized patient cohort with COVID-19 as a convenience sample[21]. All CXRs were obtained as Digital Imaging and Communications in Medicine (DICOM) images, using pydicom [https://pydicom.github.io]. We utilized frontal posteroanterior CXRs, with a separate classifier in the imaging pipeline utilized to ensure the correct orientation of the radiograph. The Python Image Library (PIL) was used to resize to a resolution of 384 × 384, 8-bit single channel [https://github.com/python-pillow/Pillow]. Images were then converted to a numpy array. The training was performed on a Linux (Ubuntu 18.04; bionic, London, England) server with Nvidia Tesla T4 (Nvidia Corporation, Santa Clara, Calif), with CUDA 11.4 (Nvidia) for 23 epochs for approximately 72 h. All programs were run in Python (Python 3.6; Python Software Foundation, Wilmington, Del) and PyTorch (version 1.01; pytorch.org). The ResNet34 CNN weights were initialized randomly and trained using a batch size of 128. AdamW was selected for optimization with an initial learning rate of 1e-3 followed by a decrease by a factor of 2 after the loss plateaued for 10 iterations with a minimal threshold of 1e-5[37]. Binary cross-entropy was used as the objective function for HCC classes, and mean squared error for age, HbA1c, BMI, and risk adjustment factor (RAF) classes. The RAF score, also called the Medicare risk adjustment, represents the amalgamation of the HCCs for a patient, and as with the HCC codes the most recent value was utilized as of December 2021[38].

Data augmentation of images was performed with random horizontal flips (20% probability), random rotations (±10 degrees), random perspective distortion of 0.2, random brightness and contrast (range 0.8, 1.2). Images were normalized with the mean (0.5500) and SD (0.1885) of the pixel values computed over the training set. Positive pixel-based occlusion-based attribution maps were generated, using the Python library Captum 0.3.1, in which areas of the image are occluded and then used to quantify how the model's prediction changes for each class [Captum model interpretability for pytorch https://captum.ai/]. We used a standard sliding window of size 15 × 15 with a stride of 8 in both image dimensions. At each location, the image is occluded with a baseline value of 0. This technique for occlusion maps does not alter the DL model and is only for visualization. Further model explanation was performed using an animated technique called "gifsplanation"[25]. This technique uses an autoencoder to modify the input image, exaggerating or curtailing certain features to show how the model's prediction changes when the input image is modified.

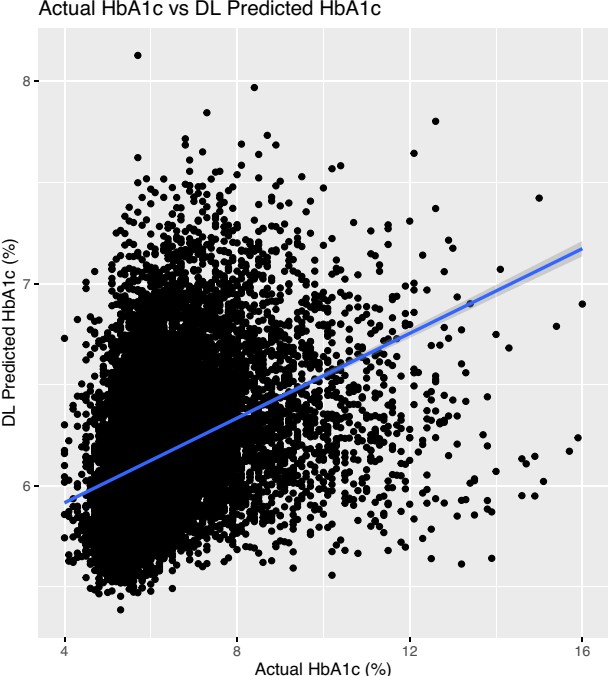

Actual HbA1c vs DL Predicted HbA1c

**Fig. 7 | Scatterplot depicting the association between actual and predicted HbA1c values using the deep learning model.** For the retrospective cohort of patients from 2010 to 2021, HbA1c values were collected within a ±30-day window of the CXRs (n = 15,945). The x-axis represents actual HbA1c values, while the y-axis shows predicted values from the deep learning model. A solid line demonstrates the linear regression fit, yielding a regression equation of $y = 0.105x + 5.497$, an $R^2$ of 0.15, and a P value < 0.001. A two-sided linear regression test was conducted, and the 95% confidence interval is displayed as a light gray band. Source data are provided as a Source Data file.

Additionally, linear regression was used to assess the model's ability to predict HbA1c compared to actual HbA1c values within a ±30-day window of the CXRs in the retrospective k-fold cohort.

As the development training dataset was assembled over a multi-year period, there were a variable number of CXRs associated with each patient ID. To avoid confounders associated with duplication, de-duplication was performed such that in the prospective, retrospective, and external validation test sets, there is a 1:1 mapping between CXR and individual patients. Therefore, for the purposes of this study, each individual CXR in both the test and external validation set represents a single patient.

### Test cohort
For the prospective test cohort, data on all patients with a frontal CXR between 1/1/2022 and 7/31/2022 was extracted. Patients who had CXRs in the development and training dataset were excluded (N = 8272) for a final total N of 9943 (Fig. 1). As done in other studies, patients (N = 136) without a diagnosis of T2D or prediabetes having two or fewer claims for evaluation and management codes (CPT codes 99202 to 99499) and five or fewer unique encounter dates were excluded[39,40], as these patients may have received care within another health system. Patients with (N = 188) diagnosis codes for type 1 diabetes (ICD9: 250.x1, 250.x3, and ICD10: E10.x) and gestational diabetes (ICD9: 648.80–648.84, ICD10: O24.4x) were removed (Fig. 1) as potential confounders[40,41].

### k-fold internal validation
Our test dataset contained 7 months of prospective (most recent) data. We wanted to see if the model was stable over time periods other than the most recent (prospective) one as a sanity check. First, we held aside the N = 9943 prospective test set. We then went back

to the original training set (N = 271,065) and split the dataset into five equal parts. The model was then retrained on 4/5ths of the original training set (with a similar 90%/10% train/validation split) and the missing 1/5th fold was used as our 'out of fold' test set. This was repeated for the five combinations, maintaining constant hyperparameters, and patient grouping rules identical to those described above. The CXRs were organized into groups based on the patients' IDs to prevent them from being divided between the training and validation sets. The same 1:1 mapping between CXR and individual patients was performed to create this retrospective dataset. We emphasize that at no time did the prospective data or external data bleed into this accessory analysis. Patient characteristics are listed in Supplementary Table 2 and performance results in Supplementary Table 3.

### External validation
A valid criticism of DL models is that they perform on their in-sample subject population but quickly fail on out-of-sample data. Therefore, we sent the model to a different academic medical center in a geographically different area. This inference utilized N = 5026 CXRs, without model calibration. The external validation data were also conventional frontal CXRs, obtained between 2019 and 2020 (Fig. 1), with ground-truth labels for T2D as described in outcome ascertainment. Results are shown in Supplementary Tables 5–7.

### Incidence detection of T2D
In both the external validation cohort and the retrospective k-fold cohort, we calculated the incidence rate of T2D using the earliest available CXR as the index date and diagnosis date for T2D. Furthermore, in the 11-year retrospective k-fold cohort, we computed the time-dependent area under the receiver operating characteristic curve at 1, 3, 5, and 10 years using a nearest neighbor estimation[42]. A left-truncation of the retrospective k-fold cohort data (excluding patients with the diagnosis of T2D before CXR) with the earliest CXR data representing the index date was performed. The time-to-event was calculated as the difference between the CXR index date and the earliest T2D diagnosis or last patient encounter date, with times censored at 7/31/2022.

### Model implementation
The model was deployed utilizing an Nvidia triton inference server (Nvidia Corporation, Santa Clara, CA). The radiology information service (Epic Radiant) was used to identify patients with CXRs, which were then written to a SQL database at regular intervals. The inference server performs a timed query to the SQL database to obtain a list of accession numbers several times a day, which were subsequently batch processed for image transfer to the server on a regular interval. The inference predictions were then written back to the SQL database.

For each patient the following features were also extracted from the EHR: age, self-reported sex, zip code, self-reported race and ethnicity, language preference, and BMI (kg/m²). Data was retrieved from the Data Warehouse using structure query language through the SQL Server Management Studio software (Microsoft, version 18.5; Redmond, WA).

Because T2D is strongly associated with geographic health inequities, we imputed the publicly available SDI by referencing the associated zip-code tabulation areas, and added it as a covariate in the LR models[43]. The SDI is based on the American Community Survey and is used to "quantify levels of disadvantage across small areas, evaluate their associations with health outcomes, and address health inequities"[44]. SDI is a metric that combines demographic data of poverty, high school dropouts, renting, overcrowding, lack of car ownership, and unemployment into a granular zip-code-level ranking. SDI, together with other measures, can help identify areas that may need additional healthcare resources.

### LR model and CXR added DL model comparisons and subgroup analysis

Binomial LR models were used to compare the performance of the CXR DL model to the model without the CXR. LR models used the diagnosis for T2D as the dependent variable, with the independent predictors of patient age, sex, self-reported race and ethnicity, self-reported language preference, BMI, SDI with or without the CXR DL prediction. The patient age at the time of CXR, as well as the most recent available BMI, were used. Self-reported race and ethnicity was normalized to three categories, non-white, white, and unknown, and similarly self-reported language preference was also normalized to non-english, english, and unknown. Four scenarios were developed to measure the models' ability to predict: patients with (1) T2D, (2) poorly controlled diabetes, (3) T2D in cohort with a BMI <25 and age 35–70 years, and (4) T2D in cohort with a BMI ≥25, age 35–70 years. The purpose of these subgroup analyses is to assess the models ability to discriminate between patients with and without T2D, with and without poorly controlled T2D, T2D patients who should be screened according to the USPSTF guidelines and patients with BMI <25 who would not undergo typical routine screening. In each case the method of Delong[45] was used to compare receiver operating characteristic (ROC) curves between the models with and without the DL prediction, with the R "pROC" library, which extends the method for unpaired comparisons.

### Sample size calculation

For the prospective cohort, a power calculation was performed a priori, and to achieve 80% power or higher a predetermined estimated sample of 8,452 DL predictions was required, assuming 15% prevalence at an alpha of 5%. Thus, with 9943 DL predictions and a prevalence of 16%, our study achieved our minimum threshold of 80% power.

### Statistical analysis

Characteristics were described using means and SDs for continuous features and percentages for categorical variables. The CXR T2D diagnostic score from the model ranged from 0 to 1, indicating the probability of T2D. The two-sided Wilcoxon rank sum test was used to assess the differences between the T2D CXR DL model score and T2D. The Kruskal–Wallis test was used to evaluate differences in the T2D CXR DL model score and T2D disease for the following groups: no T2D, controlled T2D, and poorly controlled T2D.

Proportions were tested with two-sided chi-square; means were tested with two-sided $t$-test. $P < 0.05$ was considered to indicate a statistically significant difference. ROC AUCs, 95% CIs and comparisons were calculated by the method of Delong[45]. Positive predictive value, sensitivity, specificity, and F1 scores were calculated for each cohort. Youden's J index, also known as Markedness or deltaP, was used to identify the optimal threshold for classification prediction for reporting precision, recall, and F1 scores. The optimal threshold for the deep learning predictor using this method in both prospective and retrospective datasets ranged from 0.04 to 0.16. In the model of all cases of T2D versus all other controls, the threshold was 0.1 for both the prospective and retrospective datasets. However, for the external validation dataset, the optimal threshold was 0.20. To assess equity, model performance was evaluated across self-reported race and sex (when available) from the EHR (Table 3). For multiple comparisons of ROC AUCs the Holm–Bonferroni correction was used[46]. In this study, we performed all analyses using R software (version 4.0; R Foundation for Statistical Computing, Vienna, Austria), incorporating the following packages: "survival" (version 3.2.13), "survivalROC" (version 1.0.3), and "pROC" (version 1.18.0). The Checklist for Artificial Intelligence in Medical Imaging was used for reporting in this study[47].

### Reporting summary

Further information on research design is available in the Nature Portfolio Reporting Summary linked to this article.

## Data availability

All data supporting the findings described in this manuscript are available in the article and in the Supplementary Information and from the corresponding author upon request. The model weights data are available under restricted access due to privacy and ethical considerations, because of the model's capacity to consistently predict multiple potentially identifiable comorbidities and patient age across CXRs, access can be obtained by contacting A.P., who will provide a response to inquiries within 14 days and supply necessary data use agreements. Researchers from established research institutions can request access, with the data use agreement stipulating that commercial use is not permitted. The Emory dataset (EMX) can be requested from J.W.Gichoya, who will provide a response to inquiries within 14 days, subject to a data use agreement for non-commercial use. Both internal and external validation datasets, inclusive of CXR images and select ICD10 labels, can be procured from their respective institutions to facilitate experimental replication. Source data are provided with this paper.

## Code availability

The code used in this study is freely available in Zenodo with the identifier [https://doi.org/10.5281/zenodo.7990430][48]. The license of use for the code is Creative Commons 4.0, which allows for sharing, adapting, and using the code for any purpose as long as proper attribution is given to the original authors. There are no restrictions on the availability or use of the code, and interested researchers are encouraged to download and use it for their own projects.

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

## Acknowledgements

A.P., N.S., J.W.Gichoya and S.K. received funding from the U.S. Department of Health & Human Services | NIH | National Institute of Biomedical Imaging and Bioengineering (NIBIB) - 75N92020C00008 and U.S. Department of Health & Human Services | NIH | National Institute of Biomedical Imaging and Bioengineering (NIBIB) - 75N92020C00021. B.T.L received funding from the NIH (2R01 DK104927), Veterans Affairs (1I01BX00382-01A1) and Discovery Partnership Institute (DPI), through a Discovery Partners Institute (DPI) Science Team Seed Grant Program, where the DPI is part of the University of Illinois System. J.W.Garrett and P.P. received funding from the U.S. Department of Health & Human Services | NIH |U.S. National Library of Medicine (NLM) - R01LM013151. S.K. is funded by MIDRC, NSF III 2046795, IIS 1909577, CCF 1934986 and the Alfred P. Sloan Foundation. J.W.Gichoya is funded by US National Science Foundation (grant number 1928481) from the Division of Electrical, Communication & Cyber Systems and Emerging Issues, Health Disparities; and Debasing Image-Based AI Models for Population Health (EIHD2204). Michael J. Choe, MD, Monica Harrington and Samantha Baugus, Ph.D., for her valuable editing and feedback on the manuscript. The study was primarily funded by MIDRC. The funders of the study had no role in study design, data collection, data analysis, data interpretation or writing of the report.

## Author contributions

A.P.: conceived of the presented idea, supervised the project, collected data, and performed the computations, These authors contributed equally. S.M.B.: supervised the project and co-authored the manuscript, These authors contributed equally. R.M.: network engineer, server implementation for prospective results, performed computational analysis. Z.Z.: performed external validation and computational analysis,

verified analytical methods. KT: performed external validation and computational analysis. B.P.: performed computational analysis, created figures, and contributed to manuscript writing. E.G.: contributed to manuscript writing, data analysis. N.S.: supervised the project, manuscript preparation. M.W.: supervised the project, extracted data. I.S.: lead programmer, supervised the project and performed computational analysis. J.H.S.: contributed to manuscript writing and performed computational analysis. J.M.H.: contributed to manuscript writing. P.N.: contributed to manuscript writing. M.P.L.: concept, analysis, and contributed to manuscript writing. J.M.R.F.: created figures and contributed to manuscript writing. J.W.Gichoya: external validation, performed computational analysis and contributed to manuscript writing. S.K.: programming, model design, original concept, supervised the project and contributed to manuscript writing. A.E.F.: contributed to manuscript writing, advised project. N.K.: contributed to manuscript writing, advised project. A.G.: contributed to manuscript writing, advised project. J.W.Garrett: advised project and contributed to manuscript writing. J.P.C.: developed "gifs-planation" technique and verified analytical methods. B.T.L.: contributed to manuscript writing, verified manuscript, contributed to endocrinologic assessment. P.P.: helped supervise the project and verified analytical methods. WG: helped supervise the project and verified analytical methods.

## Competing interests

A.P., S.K., N.S. are co-inventors of the patent "comorbidity prediction from radiology images," which protects the potential uses of comorbidity prediction from radiographs in value-based healthcare (applicants: DuPage Medical Group, University of Illinois, inventors: Oluwasanmi Koyejo, Andrew Chen, Patrick Cole, Nasir Siddiqui, Ayis Pyrros, U.S. Patent Application No. 17/861,347). A.P. is an advisor to Brainnet and Inference Analytics. M.P.L. is employed by Microsoft. N.K. is employed by BunkerHill Health. R.M. is an operating advisor at Ares Private Equity and CEO of Brainnet. P.P. is an advisor to Nanox-X, Bracco, and GE Healthcare. The remaining authors declare no competing interests.
