## [Peer Review File · Nature Communications]

Reviewer comments, first round –

Reviewer #1 (Remarks to the Author):

The MS has been improved. Especially the explainable-AI approach presents novel aspects convincingly suggesting some plausible features of the CXR images that can be associated with diabetes detection.

Despite the rebuttal of my suggestion to present the correlation of the DL-score and HbA1c, I would like to see these data in the MS. The explanation here ("binge eating", "binge weight loss" explaining fluctuations in HbA1c) is not appropriate, in turn many patients have consistent HbA1c levels over time. Presence or absence of a fair association can provide insights how much glycemia or rather adiposity features relate to diabetes detection from CXR images.

The speculation on why rib attenuation could be associated with diabetes detection (osteoporosis) should be taken cautiously. The correlation of (type 2) diabetes with osteoporosis is subtle and complicated, higher BMI rather inversely correlates with osteoporosis. I think that it's more probable here that rib attenuation results from increased subcutaneous fat.

Reviewer #2 (Remarks to the Author):

Please find comments regarding the response to Reviewer2 in the attached document.

Reviewer #2:

Remarks to the Author:

The authors developed and validated a deep learning model to identify persons with type 2 diabetes using deep learning. The model was developed in 271,065 chest x-rays from the authors' health system, then tested for prediction of diabetes from 9,943 chest x-rays over 7 months. The AUC was 0.84, compared to a logistic regression model which had an AUC of 0.78. The model was externally validated in 5,026 patients from Emory with an AUC of 0.77. Chest x-rays are very common. A method for opportunistically screening patients for diabetes using chest x-rays would be significant.

The manuscript Results section was very difficult to read without referring to the Methods. Recommend revising for Nature format.

Overall the Results were difficult to read and could use more text for explanation and to summarize the results.

My major criticism is that it should be explained better how this would be used in clinical practice. Statistical testing and results should also be structured to highlight utility for decision making. (authors: see Paragraph #1) The authors compare to a basic logistic regression model, but no one uses this model to decide who to test for diabetes. A more meaningful clinical comparison would be to the USPSTF recommendation to screen persons aged 35-70 with a BMI ≥ 25 . How does the CXR DL model perform compared to the USPSTF recommendation in the entire population, at a clinically meaningful operating point (rather than Youden index)? If the CXR DL model was applied within the subset of non-pregnant persons who would be screened according to USPSTF, how would this affect the number of people tested/diagnosed/missed?

Thank you for this comment. First, we will field this model in the future after further validation as an EMR-driven recommendation to physicians about their patients' risk of type 2 diabetes and consider screening by a standard approach, such as an A1C test. We discuss in the manuscript that with powerful algorithms that utilize EMR data, such as the one developed, we can enhance care for our patients by predicting and recommending screening to detect disease more proactively. Published data from the USPSTF (<https://pubmed.ncbi.nlm.nih.gov/23867023/>) demonstrates that the USPSTF has low sensitivity (between 35%-44%). The USPSTF is a guideline recommended when a patient should be screened for T2D, but it does not predict the probability of diabetes. Simply put, the USPSTF is a binomial option: 0 or 1. As we stated in the manuscript, "A total of 875 (9%) cases were evaluated as suspicious for T2D by the algorithm with no diagnosis of T2D or prior HbA1C. External validation was performed at a geographically unrelated, distinct institution (N = 5,026) with a ROC AUC of 0.77 from 2019 to 2020, where (5%) of patients were subsequently diagnosed with T2D." We discuss the utility of using CXR data for T2D screening and how this might improve population health. Based on our present study, we suggest contacting a PCP when a patient without prediabetes or diabetes has a CXR DL score higher than a defined threshold and no recent A1C screen. Future work may find additional value for the prediction score, specifically in value-based healthcare systems.

It should be more clearly stated in the manuscript when the diagnosis of diabetes was made. In other words, were patients with known diabetes (i.e. ICD code for DM or HbA1c before the CXR) excluded? For patients diagnosed after CXR, how long after the CXR was the diagnosis made? This is important, as if the diagnosis of diabetes is already known prior to the CXR, there is limited usefulness. Throughout the manuscript the authors describe "controlled" versus "uncontrolled" diabetes – does this mean that in the "controlled" cases diabetes was known and controlled by

treatment or diet/exercise? The authors should justify their presentation of odds ratios rather than hazard ratios for Figure 2 and Table 2.

Thank you, all the above information is already stated in the manuscript, under the methods section.

- In both the external validation cohort and the retrospective k-fold cohort, we calculated the incidence rate of T2D using the earliest available CXR as the index date and diagnosis code date for T2D or HbA1C date if $\geq 6.5\%$, whichever was earlier. A left-truncation of the k-fold cohort data (excluding patients with the diagnosis of T2D before CXR) with the earliest CXR data representing the index date was performed. The time-to-event was calculated as the difference between the CXR index and the earliest date of T2D diagnosis based on ICD9/ICD10 or HbA1C values $\geq 6.5\%$, with times censored at 7/31/2022.

- Furthermore, it was stated in the results:

- “In the 11-year retrospective k-fold cohort, 7,409 (25%) of the 29,420 patients had a diagnosis of T2D after the initial CXR. The incidence rate of T2D in this population was 91 (95% CI: 91, 91.5) cases per 1,000 people per year at risk.”

- In order to perform a time-dependent ROC curve, we only excluded patients in the k-fold cohort; this is referred to as a left truncation. As stated in Extended Data Table 4, 7,251 subjects did not have a diagnosis of T2D before the CXR during an 11-year period.

This is important, as if the diagnosis of diabetes is already known prior to the CXR, there is limited usefulness.

- We disagree; even if the diagnosis of T2D is known, a stratified predictor may help identify higher-risk patients, as we state in the discussion. In addition, such predictors are useful in population health and value-based programs as a method of understanding overall risk.

Throughout the manuscript the authors describe “controlled” versus “uncontrolled” diabetes – does this mean that in the “controlled” cases diabetes was known and controlled by treatment or diet/exercise?

This is already stated in the manuscript:

- The outcome of poorly controlled T2D diabetes was defined as an HbA1C value $\geq 9\%$ at any time point, as per prior studies

- Other studies have called this poor glycemic control.

- Per other references: “The threshold of 9% was chosen because, although some clinical guidelines choose 7% as the most common measure of “good” glycemic control (particularly for younger patients), and debate continues as to whether the goal of 8% is more reasonable for older persons or those with multiple comorbidities who may have a heightened risk of hypoglycemia,^{36–38} many health care systems use 9% as a generally acceptable threshold for having “poor” glycemic control, and national health care quality metrics for payment.”

[<https://pubmed.ncbi.nlm.nih.gov/31268954/>]

- As these are logistic regression models, hazard ratios do not make sense. This was an observational study; we did not treat the patients. “The hazard ratio is an estimate of the ratio of the hazard rate in the treated versus the control group.”

[<https://www.ncbi.nlm.nih.gov/pmc/articles/PMC478551/>]

- We have replaced the term “uncontrolled” with “poorly controlled” as we feel it is more descriptive.

In the first sentence of the Discussion, the study is described as a “prospective observational multisite study.” Only one of the test cohorts was acquired prospectively for 7 months in 2022 at Duly Health and Care near Chicago.

While CXRs are described as coming from 28 different locations, they are still from the same health system. It was unclear whether the data collection was truly prospective, versus whether the study was planned prospectively for the future, then chest x-rays were collected retrospectively after the fact. The external validation EMX dataset from Emory was retrospectively analyzed. It is arguable whether the overall study can be called a “prospective study” given this design.

Thank you, we have accordingly changed the title. However, to reiterate: the server and model were run prospectively, with an additional retrospective analysis done.

Please explicitly describe whether any patients in the prospective test cohort were also included in the development dataset. Currently the Cohort section of the Methods says that “all patients with a frontal CXR between 1/1/2022 and 7/31/2022”, though if taken verbatim suggests that a patient in the development dataset who later has a CXR in the first half of 2022 would be in both development and test datasets.

Respectfully, this is shown in Figure 1, which is referenced in the sentences quoted above. Those 8,272 patients who had data in training/development were excluded. However, we have added additional language to further clarify this point.

In Extended Data Table 3, radiographs are described as “cases and controls.” Was this a case control design, or a cross sectional cohort design? From my reading of the Methods I think it is the latter, but the Data Table suggests the former. If this was a case control design, then the sensitivity, specificity, and other reported results are not meaningful.

Respectfully, this is not a case-control design; it is an observational study. Additionally, this is already stated in the manuscript:

- Negative controls, henceforth referred to as controls, were chosen to mimic the deployment environment, and because the DL model would be deployed on all adult CXRs, controls, including all available CXRs in the date range above, were utilized, along with cases of T2D.

For extended data table 3, the comparison for the p value should be stated (i.e. is it a comparison of AUC, prevalence, PPV, etc?).

Thank you, we have added footnotes to tables.

Several comments on Table 2:

1. For Table 2, please state which comparison the p value refers to. It is difficult to compare AUCs, as the populations included in the two comparator groups appear to be slightly different (e.g. in Table 2 rows 1 and 2, $1561+8382 = 9943$ for CXR DL and $1554+8126=9680$ for the clinical LR model).

o Respectfully, this is written on the page, please see the footnote for the table, which reads as follows: *LR models all included: age, sex, BMI, race/ethnicity, language preference, SDI; differences in case counts between DL and LR models are due to observations being deleted due to missingness in LR models.

o In addition, AUCs can be compared with the DeLong method, and do not require exactly equal numbers of patients; paired or unpaired comparisons are possible.

■ Comparing AUCs of Machine Learning Models with DeLong's Test – Glass Box (glassboxmedicine.com)

2. In Table 2, the Youden index is used to calculate the operating point for PPV/NPV/sensitivity/specificity. It is difficult to compare between two tests with this methodology, as two separate operating points were

This comment was cut off, but we presume that the reviewer meant to say used. The Youden Index between these multiple models was similar, but many deep learning papers use the Youden Index to generate sensitivity and specificity. Youden Index method reference:

<https://pubmed.ncbi.nlm.nih.gov/16161804/>

3. For Table 2, can the authors explain why they did not include NPV to round out sensitivity/specificity/PPV?

We initially started with precision and recall and added specificity per the editor's request. The issue becomes more about space; models such as this tend to have higher NPV given the imbalanced nature of the data, in addition, clinically, PPV and sensitivity are of more interest.

4. For Table 2, please explain why the prevalence of DM2 is only 10.8% in the Row 4 Clinical LR row. I believe it should be $885 / (885+3909) = 18.5\%$ and that 10.8% is a typo.

Yes, thank you, that is a typographical error and has been corrected.

5. For Table 2 row 2, it is unclear what are the cases and controls. Does the "all cases" refer to all of the patients included in the study, or all of the cases with DM2? It is confusing because I think "cases" has a different meaning than in the "cases and controls" column.

Yes, we clarified the meaning in the table row.

Likewise, the naming of columns in Extended data table 4 needs more explanation. Are "cases" patients with incident DM2? Survivors survival free of DM2?

Yes, cases, as stated in the manuscript, mean positive cases of T2D. We have added a label in the table; yes, survivors have no T2D at that time point.

The code and model are available only by "reasonable request" and with restrictions. Suggest reviewing Nature policy: <https://www.nature.com/nature-portfolio/editorial-policies/reporting-standards>

Thank you, we have reviewed them.

Reviewer 1 Report

1. Reviewer 1: The MS has been improved. Especially the explainable-AI approach presents novel aspects convincingly suggesting some plausible features of the CXR images that can be associated with diabetes detection.
 - a. *Response: Thank you; we appreciate the comment.*
2. Despite the rebuttal of my suggestion to present the correlation of the DL-score and HbA1c, I would like to see these data in the MS. The explanation here ("binge eating", "binge weight loss" explaining fluctuations in HbA1c) is not appropriate, in turn many patients have consistent HbA1c levels over time. Presence of absence of a fair association can provide insights how much glycemia or rather adiposity features relate to diabetes detection from CXR images.
 - a. *Response: Thank you; we understand your point and have included an analysis of the correlation between the DL predicted HbA1c and actual HbA1c. For the retrospective cohort of patients between 2010 and 2021, we collected all HbA1c values within a +/-30-day window of the chest radiographs (n=15,945) and conducted a linear regression analysis between the HbA1c predicted by the DL model and the actual HbA1c values. The results showed a statistically significant linear regression model ($P < 0.001$) with an R^2 value of 0.15, indicating a relatively low correlation between the DL-predicted HbA1c and actual HbA1c values. This highlights the fact that the DL model heavily relies on adiposity features, which may not always correspond to an accurate HbA1c measurement near the time of the chest radiograph. Thus, the DL prediction cannot replace traditional HbA1c measurements. The data from the prospective cohort obtained in 2022 was not included in this analysis due to limited data within the window of time (n=47 HbA1c values); likewise, the external cohort also lacked sufficient data. We hope this information provides a better understanding of the relationship between the DL-predicted HbA1c and actual HbA1c and how adiposity features may affect the DL model's ability to predict A1C.*
3. Reviewer 1: The speculation on why rib attenuation could be associated with diabetes detection (osteoporosis) should be taken cautiously. The correlation of (type 2) diabetes with osteoporosis is subtle and complicated, higher BMI rather inversely correlates with osteoporosis. I think that it's more probable here that rib attenuation results from increased subcutaneous fat.
 - a. *Response: Thank you; we agree with your point and have rephrased the sentence in the discussion accordingly:*
 - i. *“The attenuation of the ribs and clavicle can increase the DL prediction, which could represent an increasing amount of adiposity obscuring the osseous structures, with other possibilities including age and diabetes-related osteoporosis²⁸.”*

Reviewer #2 and Reviewer #3

1. Reviewer #3: The manuscript Results section was very difficult to read without referring to the Methods. Recommend revising for Nature format. Overall the Results were difficult to read and could use more text for explanation and to summarize the results. Reviewer #3: This comment has not been addressed – the authors should clarify whether they took it into consideration and if so specify what changes were made. As above, the authors have not stated whether/how this is addressed
 - a. Response: Thank you; we apologize and were under the impression that the original manuscript would be made available to the next reviewer. We extensively rewrote and restructured the results section from the original manuscript to better align with Nature's guidelines and improve readability. We are providing all of this below in the interest of full disclosure; these are the highlighted changes that were made:

Results

Dataset summary. Our DL model trained on was developed from 271,065 CXRs from an 11-year period (160,244 unique patients), sourced from 2010 to 2021 (the training or development cohort, our training dataset), which was first prospectively tested/evaluated on 9,943 CXRs in 2022 (the prospective cohort, our test cohort) and then dataset (Fig 1). The original training dataset was further evaluated by k-fold techniques, the retrospective internal validation dataset. We next externally validated on with 5,026 CXRs (Fig. 1) from a separate institution (the Emory cohort, our external validation cohort). The training set of 271,065 dataset.

Main analysis results. We developed a deep learning (DL) model using 11 years of data from 160,244 patients using their first ambulatory CXR to produce a prediction of the diagnosis of DM2 (Extended Data Table 1). We also produced a logistic regression (LR) model that did not include any image information from the CXRs was then further analyzed synthetically by k-fold techniques (the synthetic cohort).

The prospective performance of the CXR DL model for the prediction of T2D in a separate test cohort (Fig. 1) included of 9,943 patients, most of whom had no T2DM (n = 8382, 71.9%), followed by controlled T2DM (n = 1,119, 11.3%) and uncontrolled T2DM (n = 442, 4.4%) (Table 1). Patients with T2DM tended to be older than those in other cohorts at 67 years (SD: 12.7), and there was a predominance of female patients in the nondiabetic cohort, but male patients in the controlled T2DM and uncontrolled T2DM cohorts. Regarding race/ethnicity, White Non-Hispanic individuals were prevalent in each subgroup, followed by Hispanic; Asian, Non-Hispanic; and Black, Non-Hispanic individuals. In addition, patients with uncontrolled T2DM had higher BMI and social deprivation index (SDI).

The main analysis compared model performance for predicting all subjects with T2DM (area under the curve [AUC] = a CXR was 0.84, (95% confidence interval [CI]: 0.83, 0.85), which was significantly better than) compared with the clinical logistical regression (LR) model (, which had an AUC = of 0.73, (95% CI: 0.72, 0.75; P < 0.001), as shown in Table 2. In for comparison of the significance of the AUC difference between the two models.)

In subjects with uncontrolled T2DM/poorly controlled T2D (defined as HbA1c $\geq 9\%$) $>9\%$ at any time for a patient) versus all other subjects (including those with and without T2DM), the DL others, the CXR DL predictor demonstrated a similar performance (AUC = 0.8485, 95% CI: 0.83, 0.86), which was better than that). In a subgroup analysis of subjects who meet the comparative LR model (USPSTF criteria in screening² for T2D (BMI ≥ 25 , age between 35 and 70 years), the CXR DL predictor had an AUC = 0.74, 80 (95% CI: 0.7179, 0.75; P < 0.001). In 82), and in a cohort for which BMI < 25 (ages between 35 and 70), the CXR DL model (reached an AUC = 0.89, (95% CI: 0.85, 0.93) also). The CXR DL model consistently outperformed the clinical LR model (AUC = 0.76, 95% CI: 0.71, 0.81; P < 0.001). In subjects who meet the USPSTF criteria in screening for T2DM (BMI

~~≥ 25 , age between 35 and 70 years), the DL predictor (AUC = 0.80, 95% CI: 0.79, 0.82) outperformed the LR model (AUC = 0.73, 95% CI: 0.71, 0.75). at a significance level of <0.001 . Full results are presented in Table 2.~~

~~When~~

~~To evaluate the DL score was used importance of the CXR DL's prediction overall, we added it as an additional predictor in the input into an LR model, (DL with LR model). The CXR DL predictor contribution dominated the overall performance LR via its odds ratio (Figure 2, Table 2). AUCs, for the prediction of DM2, improved in all four scenarios, with the DL predictor having vs. the largest odds ratios (Fig. 2, Table 3). The combined DL model performance for subjects with T2DM versus all cases had an AUC of clinical LR baseline model; however, it was not statistically significant, 0.85 (95% CI: 0.84, 0.85; Fig. 2), with the DL predictor having an OR of 96) versus 0.84 (95% confidence interval [CI: 66,136;]: 0.83, 0.85, $P < 0.001$). Likewise, the DL model performance for subjects with uncontrolled T2DM versus all cases was =0.16), and improved (AUC = 0.85, 95% CI: 0.83, 0.87), as was the DL model performance for subjects with T2DM versus subjects for the subset of patients who meet the met USPSTF criteria in screening for T2DM criteria (AUC = 0.81, 95% CI: 0.90, 0.82), and the DL model performance for subjects with T2DM versus BMI <25 (ages between 35 and 70) (AUC = 0.88, 95% CI: 0.83, 0.92).80, 0.83), also included in Table 2.~~

~~In the evaluation of CXR T2DM As shown in Figure 3, DL model predictions, for subjects with T2DM T2D had significantly higher predictions than subjects without T2DM T2D (median 0.28; interquartile range [IQR]: 0.15, 0.49 vs. median 0.04; IQR: 0.00, 0.14; $P < 0.001$; Fig. 3A). Subjects with uncontrolled T2DM poorly controlled T2D had higher scores (median 0.35; IQR: 0.20, 0.58) than subjects with T2DM T2D (median 0.26; IQR: 0.0, 0.45) or no diabetes (median 0.03; IQR: 0.0, 0.11; $P < 0.001$; Fig. 3B). Occlusion maps were generated to display the basis for model decision (Fig. 4), with features corresponding to the central chest, lower neck, upper abdomen and axillary regions, as areas of visceral fat deposition. 3B).~~

~~Results In the prospective test cohort, among all ages, 1,370 (14%) patients were identified by the model as high risk using Youden's Index²² who did not have an HbA1c value or a diagnosis of T2D, representing potential screening opportunities. Of these, 58 would not have met the criteria for screening per the USPSTF² (BMI less than 25 amongst all ages), with an additional 44 uncertain based on no available BMI.~~

~~Retrospective (k-fold validations on the full 11-year development dataset to create a synthetic cohort (out-of) Validation Results. Internal retrospective validation was performed as specified in k-fold predictions) demonstrated internal validation in methods (Extended Data Table 2). Results were similar performance to the prospective test set, with ROC the DL model producing an AUC of T2DM with CXR DL yielding 0.83 (95% CI: 0.82, 0.83) versus 0.85 (95% CI: 0.84, 0.85) for in the prospective dataset internal test cohort (Extended Data Table 3). As in the prospective cohort In subjects with poorly controlled T2D versus all others, the DL predictor outperformed (AUC = 0.83, 95% CI: 0.82, 0.83; $P < 0.001$) the clinical LR model (AUC = 0.79, 95% CI: 0.78, 0.79; $P < 0.001$). In subjects with uncontrolled T2DM (defined as HbA1c $\geq 9\%$) versus all other subjects (including those with and without T2DM), the DL predictor demonstrated a similar performance (AUC = 0.82, 95% CI: 0.81, 0.82), which was better than that of the comparative LR model (AUC = 0.76, 95% CI: 0.75, 0.76; $P < 0.001$). In a cohort for which BMI was <25 , the CXR DL model (AUC = 0.83, 95% CI: 0.81, 0.84) also outperformed the clinical LR model (AUC = 0.78, 95% CI: 0.77, 0.79; $P < 0.001$). In subjects who meet the USPSTF criteria in screening for T2DM T2D (BMI ≥ 25 , age between 35 and 70 years), the DL predictor (model reached an AUC = 0.79, 95% CI: 0.79, 0.79) outperformed the LR model (AUC = 0.73, 95% CI: 0.73, 0.74). Results are tabulated in Extended Data Table 3.~~

In the prospective test cohort, among all patients at or above the threshold of 0.10 for the CXR DL predictor, 875 (9%) patients were flagged by the model who did not have a HbA1c value or a diagnosis of T2DM, representing potential screening opportunities. Of these, 44 patients had a BMI <25 or no BMI data available.

Incidence Detection of T2D. In the 11-year development training set retrospective k-fold cohort, 7,409 (25%) of the 29,420 patients had a diagnosis of T2DM subsequent to T2D after the initial CXR. The incidence rate of T2DM/T2D in this population was 91 (95% CI: 91, 91.5) cases per 1,000 person-years people per year at risk. Of these 7,409 patients, 5,292 (71%) had a DL prediction >0.10. Time-dependent ROC curves at 1 year (AUC = 0.80, 95% CI: 0.79, 0.80), 3 years (AUC = 0.79, 95% CI: 0.78, 0.80), 5 years (AUC = 0.79, 95% CI: 0.79, 0.79), and 10 years (AUC = 0.78, 95% CI: 0.77, 0.79) demonstrated similar performance (Extended Data Table 4), indicating a lack of change over time. Using the CXR as the index date, the delay in diagnosis was an average of 1,067 days (SD±1,000 days).

Subgroup Analysis by Race External Validation Results. The validation of the DL model was performed on an external data set of ambulatory frontal CXRs at Emory from 2019 to 2020 (Extended Data Table 5 and ~~See~~6), and we observed an AUC of 0.77 (Extended Data Table 7). In this cohort, the incidence rate of T2D was 20.4 (95% CI: 18, 23) cases per 1,000 person-years at risk, with 249 patients diagnosed with T2D after the initial CXR. Of the 249 patients, the model flagged 146 (59%) for potential earlier screening.

Demographics. Of the prospective test cohort's (Fig. 1) 9,943 patients, most had no T2D (n = 8382, 71.9%) with controlled T2D in (n = 1,119 (11.3%) and poorly controlled T2D in (n = 442 (4.4%) (Table 1). Patients with T2D tended to be older than those in other cohorts at 67 years (SD: 12.7), and there was a predominance of female patients in the nondiabetic cohort and male patients in the controlled T2D and poorly controlled T2D cohorts. Regarding race/ethnicity, white Non-Hispanic individuals were prevalent in each subgroup, followed by Hispanic; Asian, Non-Hispanic; and Black, Non-Hispanic individuals. In addition, patients with poorly controlled T2D had higher BMI and social deprivation index (SDI). Demographics in the training dataset were similar and are shown in Extended Data Table 1.

Model Equity. Previous studies have shown that convolutional neural networks can easily learn self-reported race and other sensitive attributes ~~can be easily learned by convolutional neural networks~~^{22,23} attributes^{23,24}. Out of concern that spurious features related to these sensitive attributes could be contributing to this diabetes prediction model, a subgroup analysis was conducted and shown in Table 3. ~~All subgroups demonstrated an AUC of greater than 0.80 and were not statistically different (P > 0.05).~~ Subgroup analysis failed to achieve statistical significance (P > 0.05), suggestive of a lack of bias.

External validation. External validation of the model from frontal CXRs at Emory from 2019 to 2020 demonstrated similar performance with an AUC of 0.77 (Extended Data Table 7). In the Emory external validation cohort, the incidence rate of T2DM was 20.4 (95% CI: 18, 23) cases per 1,000 person-years at risk, with 249 patients diagnosed with T2DM after the initial CXR. Of the 249 patients, 146 (59%) had a DL predictor value ≥ 0.10 .

Model explainability. Occlusion maps were generated to display the basis for the model decision (Fig. 4 and Fig 5), with image features predictive of T2D corresponding to the central chest, lower neck, upper abdomen, and axillary regions. In Figure 5, we took a random sample of 48 occlusion maps from the internal and external cohorts to demonstrate that the same features were being used. In addition, we used an autoencoder and a latent shift to generate an animation ("gifsplanation") (Figure 6), exaggerating and curtailing anatomic features used for prediction from a representative frontal radiograph²⁵. This method also does not alter the model weights and demonstrates that central fat distribution (mediastinal, upper abdomen, and supraclavicular regions), as well as attenuation of the ribs and clavicle, drives the prediction for T2D. This animation can be directly viewed at

<https://www.dropbox.com/s/ox9be7hnhvni81f/Gifsplanation-small-voice.mp4?dl=0>, with multiple randomly selected examples.

2. Reviewer 3: If blood pressure data is available then I think it should be possible to compare the USPSTF approach and see how it performs as a screen, compared with the CXR DL approach. If it is not available then I think it is appropriate to add a remark to the Discussion section that direct comparison with that approach is not possible.
 - a. *Response: The current USPSTF recommendation for screening for type 2 diabetes (<https://www.uspreventiveservicestaskforce.org/uspstf/recommendation/screening-for-prediabetes-and-type-2-diabetes>) is in non-pregnant adults between the ages of 35 and 70 years who have a BMI greater than 25 and does not include blood pressure data, which was previously included in older guidelines. As a result, a direct comparison to the model is not possible because not all patients with a BMI over 25 had an available HbA1c measurement in our cohorts. The following statement has been added to the manuscript to reflect this: "It was not possible to directly compare the USPSTF criteria in our sample due to the lack of HbA1c data in many patients with a BMI over 25."*
3. Reviewer 3: For patients diagnosed after CXR, how long after the CXR was the diagnosis made? This has not been answered by the authors.
 - a. *Response: Thank you; we believe this was already addressed in the manuscript in the Results section under "Incidence detection": "Using the CXR as the index date, the delay in diagnosis was an average of 1,057 days (SD±1,005 days)." Additionally, for further clarification we also have added the median and IQR to the sentence.*
4. Reviewer 3: Since the reviewer requested that it should be "more clearly" stated, then the authors could revise their wording to be more explicit.
5. *Response: Thank you; we have accordingly revised the above paragraph as follows:*
 - a. *In order to calculate the incidence rate of type 2 diabetes (T2D), we considered both the external validation cohort and the retrospective k-fold cohort. The earliest available CXR was used as the starting point for our calculations, serving as the index date. We also looked at either the diagnosis code date for T2D or the HbA1c date (if it was ≥6.5%), whichever came first. In the k-fold cohort, we only included patients in our analysis who did not already have a T2D diagnosis prior to their earliest CXR. The time-to-event was calculated as the elapsed time between the CXR index date and the earliest date of T2D diagnosis, which was based on either ICD9/ICD10 codes or a HbA1c value ≥6.5%. All calculations were censored at 7/31/2022.*
6. Reviewer 3: "A left-truncation of the k-fold cohort data" Is this also done for the external validation cohort?
 - a. *Response: Thank you for the comment. No, the number of patients who developed diabetes in the external cohort was too small to allow a meaningful calculation; as stated in the manuscript: "249 patients diagnosed with T2D after the initial CXR." We also have added the following sentence for further clarification: "Additional time-dependent ROCs were not performed on the external cohort, because of small sample size and length of time."*
7. Reviewer 3: The time-to-event was calculated as the difference between the CXR index and the earliest date of T2D diagnosis based on ICD9/ICD10 or HbA1c values ≥6.5%, with times censored at 7/31/2022. Is this data reported in the manuscript?
 - a. *Response: Thank you for the comment. As stated above, yes, this was stated in the manuscript: "Using the CXR as the index date, the delay in diagnosis was an average of 1,057 days (SD±1,005 days)."*
8. Review 3: Remark : In Figure 2 the y-axis has a variable called CXR DL T2DM - T2DM is not explained or mentioned elsewhere in the (searchable) text.
 - a. *Response: Thank you; we can change the figure to match the manuscript: "T2D CXR".*

9. Reviewer 3: I don't think this addresses the original remark. The authors removed the word "multisite" but not the word "prospective" from the first sentence of the Discussion. I don't think the study can be called prospective as it was not designed or intended at the time that the data was collected.
- a. *Response: Thank you for pointing this out. We think there may be some confusion as this study has several components. Reviewer #2 states "It was unclear whether the data collection was truly prospective, versus whether the study was planned prospectively for the future." To clarify this point, we specifically designed the server to run the model prospectively, meaning we collect data daily, and this was detailed under "Model implementation". We also performed a power calculation for the prospective component of this study as part of our planning. Furthermore, we have updated the paragraph titled "Model implementation" to make this hopefully more clear:*
 - i. *"The model was deployed utilizing an Nvidia triton inference server (Nvidia Corporation, Santa Clara, Calif). The radiology information service (Epic Radiant) was used to identify patients with CXRs, which were then written to a SQL database at regular intervals. The inference server performs a timed query to the SQL database to obtain a list of accession numbers several times a day, which were subsequently batch processed for image transfer to the server on a regular interval. The inference predictions were then written back to the SQL database."*
 - b. *We understand the entire study was not prospective, as we did a retrospective analysis on our development and training data and an external validation of outside retrospective data from Emory. To address these concerns, we revised the sentence and removed the word "prospective" as the study did have multiple components.*
10. Reviewer 3: Respectfully, this is shown in Figure 1, which is referenced in the sentences quoted above. Those 8,272 patients who had data in training/development were excluded. However, we have added additional language to further clarify this point. Please clarify what text has been altered and how.
- a. *Response: Yes, thank you; we added the following sentence in the Methods section under "Test cohort": "Patients who had CXRs in the development and training dataset were excluded (N=8,272) for a final total N of 9,943 (Fig. 1)."*
11. Reviewer 3: Respectfully, this is written on the page, please see the footnote for the table, which reads as follows: *LR models all included: age, sex, BMI, race/ethnicity, language preference, SDI; differences in case counts between DL and LR models are due to observations being deleted due to missingness in LR models. This does not answer the question. I do not follow which two models are compared (in each case) to achieve the p values presented.
- a. *Response: Thank you for this comment. There may be some confusion as Table 2 was updated into a horizontal layout over a stacked layout from the original table. The "P Value" column has double arrow ↔ indicating that it compares the Clinical logistic regression model and the CXR Deep Learning model, and likewise the CXR Deep Learning model and Deep Learning with Logistic Regression models. We have also added the following to the footnote in the table: "2P Value, AUC comparison using the method DeLong between the adjacent models."*
12. Reviewer 3: DeLong's original method, as per your reference 46 (and the link included here) applies only to paired comparisons. The R implementation extends it for unpaired datasets. This should be clarified in the text as it is non-standard usage. <https://search.r-project.org/CRAN/refmans/pROC/html/roc.test.html>
- a. *Response: Thank you for pointing that out; we have added it to the text: "In each case the method of DeLong⁴⁶ was used to compare receiver operating characteristic (ROC) curves between the models with and without the DL prediction, with the R "pROC" library, which extends the method for unpaired comparisons."*
13. Reviewer 3: For Table 2, can the authors explain why they did not include NPV to round out sensitivity/specificity/PPV? I do not agree with this – clinically PPV and NPV are both extremely

important in order to know how the results of a test should be interpreted. If PPV is included then NPV should be also.

- a. *Response: Yes, we understand your point, and we have added NPV to all the tables where appropriate.*
14. Reviewer 3: Likewise, the naming of columns in Extended data table 4 needs more explanation. Are “cases” patients with incident DM2? Survivors survival free of DM2?The word “survivor” is not used anywhere else in the (searchable) text, and it is not clear to me why it is used to refer to subjects without T2D. I also do not understand the word “censored” in this table. Terminology should be clear and stick to the same convention throughout the paper.
 - a. *Response: Thank you for bringing this to our attention. We view the time-dependent ROC as representing a type of survival analysis, borrowing language from there; but accordingly we have changed the column title to what it represents: Patients Not yet evaluated with a diagnosis of T2D or Lost to followup. It means those patients have no documented T2D or are lost to followup, and therefore their status is unknown/unknowable. Accordingly we have added a footnote to the table for further clarification: “*Category indicates patient is either pending physician visit without T2D diagnosis or has been lost to followup.”*
15. Reviewer 3: The code and model are available only by “reasonable request” and with restrictions. Suggest reviewing Nature policy: <https://www.nature.com/nature-portfolio/editorial-policies/reporting-standards>
This reply is insufficient. The Nature policy states that any restrictions on availability must be disclosed to the editors at the time of submission and also in the submitted manuscript. The authors need to have a clear policy on what they mean by “reasonable request” and what “restrictions” are in place, and this should be clarified to editor and reviewers.
16. *Response: Thank you for bringing this matter to our attention. We have emailed the editor and changed our data availability statement to align with Nature’s Communications statement, <https://www.nature.com/documents/nr-data-availability-statements-data-citations.pdf>:*
 - a. **Data Availability.** The model weights used in this study are only available through controlled access due to privacy concerns. Interested researchers may request access to the model weights by contacting the data custodian AP. The custodian will respond to access requests within 7-10 days and will provide the necessary data use agreements. The data use agreements will include restrictions on model use, which must be adhered to by the requester. Underlying chest radiographs are not available because of HIPAA restrictions.
 - b. **Code Availability.** The code used in this study is freely available on Github at <https://github.com/apyrros/HCC-comorbidities>. The license of use for the code is Creative Commons 4.0, which allows for sharing, adapting, and using the code for any purpose as long as proper attribution is given to the original authors. There are no restrictions on the availability or use of the code, and interested researchers are encouraged to download and use it for their own projects.

Reviewer comments, further round review –

Reviewer #1 (Remarks to the Author):

The authors have adequately addressed my points of criticism.

Reviewer #2 (Remarks to the Author):

All comments and suggestions have been addressed by the authors sufficiently from my perspective. My only remaining concern is regarding the data availability - I have several remarks on that, but I leave it to the editor to decide what is required by the journal.

- 1) Model weights will be available through a data custodian if deemed appropriate, for reasons of privacy concerns. I am not aware of any privacy concerns that apply to the sharing of model weights - I would question whether this restriction is necessary or beneficial to the progress of open science.
- 2) If the model weights are not to be made openly accessible then the limitations of access should be clearly stated e.g. available to researchers at recognized research institutes, not permitted for use in commercial development etc.
- 3) CXR images will not be available because of HIPAA restrictions: Many large databases of CXR images have been made public in recent years (Chexpert, MIMIC, Chest XRay 14 etc) in spite of HIPAA restrictions. I would suggest that such restrictions can be overcome if there is a will to share the data.

Reviewer #1 (Remarks to the Author):

The authors have adequately addressed my points of criticism.

Reviewer #2 (Remarks to the Author):

All comments and suggestions have been addressed by the authors sufficiently from my perspective. My only remaining concern is regarding the data availability - I have several remarks on that, but I leave it to the editor to decide what is required by the journal.

1. Model weights will be available through a data custodian if deemed appropriate, for reasons of privacy concerns. I am not aware of any privacy concerns that apply to the sharing of model weights - I would question whether this restriction is necessary or beneficial to the progress of open science.
 - a. *Thank you for your feedback and your valuable insights regarding data availability. We appreciate your perspective on open science and the importance of data sharing. We would like to clarify our current limitations regarding the availability of resources to address your concerns, and provide additional information regarding the privacy concerns related to the model weights.*
 - b. *In relation to the model's availability and concerns surrounding privacy, it is crucial to highlight that our model consistently generates precise age predictions and comprehensive comorbidity data (such as CHF, COPD, and DM2) for various radiographs of the same patient. The in-depth information generated by the model could function as a unique identifier, potentially exposing the patient's identity and raising privacy issues (e.g., a 95-year-old individual with CHF). Furthermore, the HIPAA safe harbor permits the inclusion of age data, except for individuals aged 89 or older, which our model can reliably predict. Regrettably, due to constraints in funding and staffing, our team is unable to guarantee the appropriate privacy compliance measures and legal assessments needed to freely distribute the model weights without limitations.*
2. If the model weights are not to be made openly accessible then the limitations of access should be clearly stated e.g. available to researchers at recognized research institutes, not permitted for use in commercial development etc.
 - a. Accordingly, we have updated our limitations of access as follows:
 - b. *We have updated the data availability statement as follows: The model weights data are available under restricted access due to privacy and ethical considerations, because of the model's capacity to consistently predict multiple potentially identifiable comorbidities and patient age across CXRs, access can be obtained by contacting AP, who will provide a response to inquiries within 14 days and supply necessary data use agreements. Researchers from established research institutes can request access, with the stipulation that commercial use is not permitted. The raw CXR and ICD10 data are protected and are not available due to data privacy laws. The Emory dataset (EMX) can be requested from JW, who will provide a response to inquiries within 14 days, subject to a data use agreement. Source data are provided with this paper.*
3. CXR images will not be available because of HIPAA restrictions: Many large databases of CXR images have been made public in recent years (Chexpert, MIMIC, Chest XRay 14 etc) in spite of HIPAA restrictions. I would suggest that such restrictions can be overcome if there is a will to share the data.
 - a. *We are aware of the publicly available CXR image databases and their success in overcoming HIPAA restrictions. However, in our case, the resources required to anonymize and process the data to comply with HIPAA regulations exceed our current capabilities. We hope to address this issue in future projects when additional resources become available. We apologize for the limitations in addressing your concerns regarding data availability. We hope to improve our capabilities in the future and contribute to the progress of open science.*